# Dimensions of Poverty in Kunduz Province of Afghanistan

Muhammad Asef Shaiq [1,2], Ali Akbar Barati [1,*] 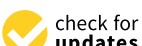, Khalil Kalantari [1] and Ali Asadi [1]

1    Department of Agricultural Management and Development, University of Tehran, Tehran 1417935840, Iran
2    Department of Agricultural Economics and Development, University of Baghlan, Baghlan 3601, Afghanistan
*    Correspondence: aabarati@ut.ac.ir; Tel.: +98-263-220-6824; Fax: +98-263-220-6825

**Abstract:** Afghanistan is a vulnerable country, and various challenges have led to widespread poverty. This study measured the different dimensions of poverty in rural and urban areas to help policymakers with poverty alleviation in the Kunduz province of Afghanistan. The data were collected from 360 rural and urban households. According to the findings, the MPI index in the Kunduz province's rural and urban areas was found to be 0.483 and 0.445, respectively. The results indicate that Kunduz faces both rural and urban poverty, but that rural poverty is more severe than urban poverty. The intensity and headcount ratio of poverty in rural areas is more significant than in urban areas. It also is clear that all dimensions of poverty in Kunduz are very serious. Thus, it is necessary to establish a comprehensive program to reduce all aspects of poverty.

**Keywords:** rural household poverty; urban poverty; intensity of poverty; poverty alleviation policies; kunduz–afghanistan

## 1. Introduction

Poverty is a worldwide concern that nations have concentrated on reducing for years [1]. Even with present improvements, the problem of poverty still exists and poses a threat to everyone, especially those who reside in rural areas. Three-quarters of impoverished people in developing countries live in rural areas [2], where inhabitants do not have access to simple hygienic facilities or clean drinking water [3,4]. Afghanistan is a vulnerable country and faces countless challenges, such as widespread poverty. Although much attention has been paid to the government and international financial support to reduce poverty, this phenomenon has increased over time [5–7].

Since 2001, the international community has made significant efforts to rebuild Afghanistan in several areas, including education and health [8], but many challenges remain. Afghanistan is primarily a rural country with a welfare gap between rural and urban areas. Rural areas make up the vast majority of the country's population, and most of these areas face poverty. In other words, four out of every five impoverished people live in rural areas, with most residing in Afghanistan's northeastern regions. Kunduz is located in this territory where poverty is increasing dramatically [9]. The rural areas of Afghanistan also face various economic, social, environmental, and physical challenges [10]. Furthermore, the poverty and housing situation between urban and rural communities is not the same, and the rural areas have worse accommodations [11]. The lack of food and housing means a low standard of living and is considered a symptom of multidimensional poverty. The Central Statistics Organization [11] has shown that the high percentage of poverty at the national level is mainly related to the people living in remote areas and villages.

It should be noted that poverty has more than one dimension and is not merely a geographical phenomenon. It needs to be defined in more detail. Numerous definitions of poverty have been put forth over the years [12,13]. As Sen [14] argued, income alone could not be merely a basic need. Recent studies [15–18] stated that the one-dimensional view of poverty reduction has not been very effective and emphasized the multidimensional aspect of poverty. Therefore, considering factors such as the ability to meet basic needs and access

to health and educational facilities, measuring the well-being of individuals is possible and effective [19].

In recent years, the Multidimensional Poverty Index (MPI) was developed to consider both the incidence of poverty and the intensity of deprivation. According to the World Bank, poverty is an "apparent deprivation in welfare" [20]. However, poverty is a multidimensional phenomenon that includes a lack of education and income, poor health status, low standard of living, quality of work and self-employment, social pressure, and discrimination. So, depending on the understanding of welfare, poverty can be defined narrowly or broadly. Many world economists complained about single-dimensional poverty. Now, there are many studies of poverty from more than one dimension. For example, in the UN Human Development Report, Alkire and Santos [21] surveyed the poverty situation in 104 countries regarding education, health, and standard of living, using the MPI. They showed that South Asia (50.9%) and sub-Saharan Africa (27.6%) are suffering from the MPI, and countries like Peru, Malawi, Sri Lanka, and Bolivia have the most privation in living standards. Iraq, Ecuador, Albania, and Guatemala have the most deprivation in terms of education, and countries such as Latvia, Hungary, and Uzbekistan have the most deprivation in terms of health. Dehury and Mohanty [22], using the Human Development Survey data of India (IHDS), estimated and analyzed the MPI dynamics in 84 natural areas. This research showed that about 50% of the Indian population faced multidimensional poverty. Barati, et al. [23,24] analyzed multidimensional poverty in Iranian rural communities. They used a database that included the 2016 census data of approximately 20,000 rural households. Based on their findings, the MPI in Iran was 0.349, and the dimensions of education, health, and standard of living have the highest share of the MPI orderly. Based on their results, the rural poverty intensity and headcount ratio were 0.558 and 0.628, respectively.

In the last two decades, various studies have been conducted in Afghanistan. The World Bank and Ministry of Economy of Afghanistan [9] studied poverty in Afghanistan and concluded that the poverty rate rose from 36% in 2012 to 40% in 2014. They also stated that half of the impoverished population is under the age of 15, about 75.6% of whom are illiterate and often live in rural areas. Trani, Kuhlberg, Cannings and Chakkal [7] showed that almost all adults in Afghanistan suffered from at least one dimension of poverty, and the poorest ethnic minorities were identified as inhabitants of rural areas, women, the elderly, and those with congenital disorders. Rahimi [25] examined the "impact of regional security and integration on poverty reduction in Afghanistan and concluded that war is an important factor and a major security problem and that instability is an important indicator of poverty in Afghanistan". He claimed that the previous government's focus on the war led to less attention on providing basic essential services such as health, education, goods, and public services. Based on the World Bank and Ministry of Economy of Afghanistan [9], which examined poverty in all provinces of Afghanistan, the contribution of the Kunduz province to the MPI was (0.43) with a headcount ratio of (0.773) and intensity of poverty of (0.556). However, the results of this study do not provide a clear understanding of rural poverty compared to urban areas, based on localized MPI indicators in the province of Kunduz.

Since each province in Afghanistan has different climatic and socio-economic situations, it is necessary to study poverty by considering these differences. Although previous surveys indicated that poverty in the rural areas of Kunduz was critical [26], a detailed and comprehensive study has not yet been completed. As stated, studying the MPI by using localized indicators is fundamental for policymakers to legislate more purposeful policies to reduce and alleviate poverty. Accordingly, to have a vigilant standpoint on the MPI in the study area and along with the relative concept of poverty, this study sought to answer the following main questions: (a) What was the difference between the poverty situation of various dimensions of poverty among rural households in Kunduz province compared to urban households? (b) Was the share of the three dimensions of poverty similar in urban and rural areas? And finally, were the intensity and incidence of poverty the same among rural and urban communities in this province?

## 2. Materials and Methods

### 2.1. Study Area

The study area was Kunduz, a province located in northeastern Afghanistan (Figure 1). This province borders the Amu Darya province in the north (300 km to Tajikistan), the Baghlan province in the south, the Takhar province in the east, the Samangan and the Balkh province in the west, and comprises the northeastern zone between the Baghlan, Takhar, and Badakhshan provinces. It covers 8080.9 km² and is divided into ten administrative units. This province consists of nine districts (Chahardareh, Aliabad, Khanabad, Imam Sahib, Dasht-e-Archi, Qaleh-e-Zal, Kalbad, Goltepe, and Aqtash) (MEA, 2019). 74% of Kunduz's population live in rural areas, and 80% of them engage in agriculture and animal husbandry. About 51% of Kunduz's population are men, and 49% are women. In terms of climate, this province is partly humid and semi-arid. Almost 75% of the Kunduz agricultural land is flat. The soil of this province is fertile, and if there is enough water, multiple products can be grown. The economic infrastructure of this province is based on agriculture and animal husbandry. Its main agricultural products are wheat, rice, barley, corn, mung bean, watermelon, melon, almond, and grape [26].

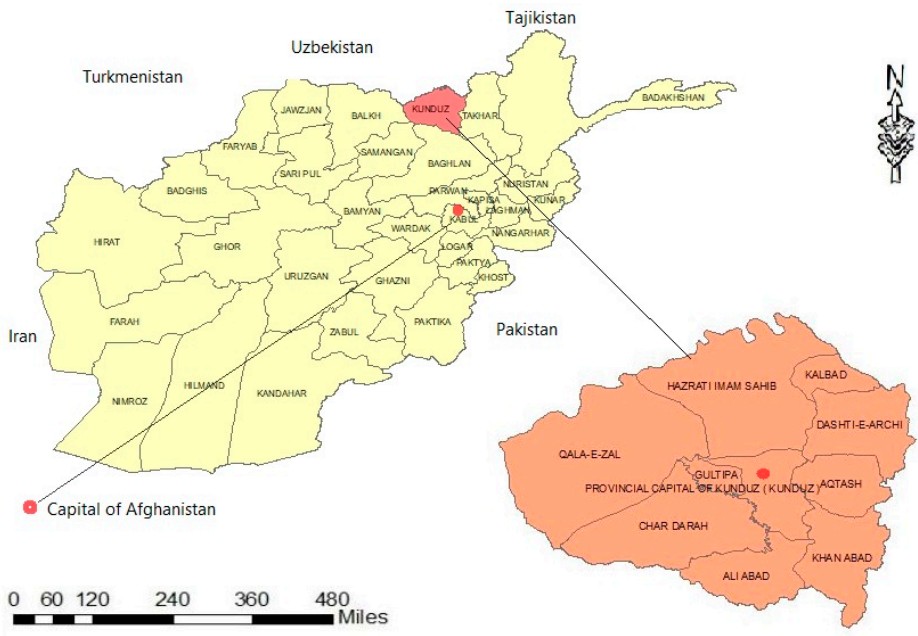

**Figure 1.** Study area location (Kunduz Province–Afghanistan).

### 2.2. Data

This research was a survey that used preliminary data. The research tool was a questionnaire that included 23 questions and indicators to evaluate the MPI [27]. The method used to collect data was field and online interviews. The sample included 360 households studied. The sample size was estimated using Cochran's method for the size of an unknown population ($n = s^2 \cdot t^2 / d^2$) because there was no reliable information on the number of households ($s = 0.145$, $t = 1.96$ and $d = 0.015$). The data collection tool was a questionnaire completed among urban and rural households of the Kunduz province. These data included variables that showed individual characteristics, educational status, health and nutrition, and the standard of living of the respondent's family.

### 2.3. Method

The MPI was used to gauge the extent of poverty in rural and urban households in Kunduz. The MPI was developed by Alkire and Foster [28] to measure the intensity of poverty and deprivation. Since poverty is a comparative phenomenon, this study attempted to change the MPI indicators to better reflect actual poverty in Kunduz. People's

perceptions of poverty are not the same in different regions and vary depending on economic, social, environmental, cultural, and additional conditions. Therefore, measuring it requires appropriate indicators specifying absolute poverty in the study population. For example, a family may be considered poor if it lacks access to electricity, but it may not be so if it lacks access to the internet or to clean water.

2.3.1. Measuring the Multidimensional Poverty Index

The MPI was proposed by Alkire and Jahan [27] for measuring poverty. Then, the United Nations Office for the Development of Humanities developed the MPI to evaluate poverty dimensions among countries. It is one of the measurements taken to modify the headcount ratio, that was proposed by Alkire and Foster [28]. This study looked at 17 variables with the three key dimensions of education, health, and living conditions to calculate the MPI (Table 1). We also localized the indicators based on the research area. In other words, we have made an effort to use the best indicators that capture the local community's poverty in each of the three categories. In brief, the education dimension includes years of schooling and school attendance; the living standards dimension consists of water, electricity, cooking fuel, and asset ownership; and the health dimension includes nutrition and child mortality. The global MPI normatively weighs each dimension equally, and within each dimension, the indicators are also equally weighted.

**Table 1.** Deprivation indicators of households and their weights.

| Dimension (Weight) | | Indicator (Weight) | Symbol | Final Weight |
|---|---|---|---|---|
| Education (1/3) | | No household member between age 6–10 goes to school (1/3) | Edu1 | (1/9) |
| | | No household member with the age of 10 or older has literacy (1/3) | Edu2 | (1/9) |
| | | No household member has a university degree or is studying at the university (1/3) | Edu3 | (1/9) |
| Health (Health and Nutrition) (1/3) | Health (1/2) | In the last ten years, there has been a family history of stillbirth (1/4) | Health1 | (1/24) |
| | | In the last ten years in the family, there has been a family history of a child death age of 2–5 years (1/4) | Health2 | (1/24) |
| | | At least one woman in the family has died of pregnancy complications (1/4) | Health3 | (1/24) |
| | | There is a disabled person in the family due to a lack of physical health (1/4) | Health4 | (1/24) |
| | Nutrition (1/2) | For the past year, the family has not been able to provide food for all its members (1/4) | Food1 | (1/24) |
| | | At present, the family does not have the financial means to provide daily food for its members (1/4) | Food2 | (1/24) |
| | | There is a child under the age of 5 who is malnourished in the family (1/4) | Food3 | (1/24) |
| | | In the family, there are people who stay hungry during the day (1/4) | Food4 | (1/24) |
| Living Standards (1/3) | | The capacity of the family home is not enough for the family members (1/6) | Liv St1 | (1/18) |
| | | No access to electricity (1/6) | Liv St2 | (1/18) |
| | | No access to the internet (1/6) | Liv St3 | (1/18) |
| | | No access to a refrigerator (1/6) | Liv St4 | (1/18) |
| | | No access to wood or gas to cook and heat a residential house (1/6) | Liv St5 | (1/18) |
| | | No access to safe drinking water, such as tap water or deep, covered wells (1/6) | Liv St6 | (1/18) |
| | | Sum Coefficients | | 1 |

2.3.2. The Process of Calculating the MPI Was as Follows

a. Calculating the household deprivation value: First, the status of the 17 indicators for each household was determined. Then, the deprivation index ($DH_i$) was calculated based on the following equation.

$$DH_i = \sum_{j=1}^{m} (w_j * d_{ij}) \ (i = 1, 2, \ldots n) \ (j = 1, 2, \ldots m) \tag{1}$$

where $DH_i$ represents the deprivation coefficient for the $i$th household, $w_j$ denotes the weight of indicator $j$, and $d_{ij}$ denotes the deprivation index of household $i$ in indicator $j$ ($j = 1, 2, \ldots , m$). Like the United Nations Multidimensional Poverty Index, impoverished people here are considered to have a $DH_i$ of more than one-third.

b. Calculation of headcount ratio of poverty ($H$): The index of poverty in society as is the ratio of the number of rural households with deprivation and the sum of the different dimensions of poverty ($Q$) to the total number of rural households ($P$). It was calculated using the following equation:

$$H = \frac{Q}{P} \tag{2}$$

Q was calculated by Equations (3) and (4), respectively:

$$Q = \sum_{i=1}^{n} (p_i * q_i) \tag{3}$$

$q_i$ is the status of household $i$ in the MPI, and $p_i$ represents the number of individuals in household $i$.

$$P = \sum_{i=1}^{n} p_i \tag{4}$$

c. Intensity of poverty at the community level (A): Equation (5) explains the intensity of poverty at the community level which is the ratio of the weighted sum of the number of households with multidimensional poverty to the total number of households with multidimensional poverty.

$$A = \frac{\sum_{i=1}^{n} (DH_i * p_i * q_i)}{\sum_{i=1}^{n} (p_i * q_i)} \tag{5}$$

d. In the next step, we obtained the multidimensional poverty index of rural households using the following equation.

$$MPI = H * A \tag{6}$$

e. In the final stage, the contribution of each dimension (education, health, and living standards to the MPI was calculated using Equation (7).

$$C_k = \frac{\sum_{i=1}^{n} \sum_{j=1}^{m} \left( w_{jk} * d_{ijk} * p_i * q_i \right)}{MPI} \ (k = E.H.L) \tag{7}$$

In Equation (7), $k$ represents the poverty dimension and is equal to each of the three dimensions of poverty ($E$, $H$, and $L$). $d_{ijk}$ represents the deprivation status of $i$'s household of $j$ index in $k$ dimension, $w_{jk}$ represents the weight of $j$ index in $k$ dimension, $q_i$ is the status of $i$ household in terms of the MPI, and $p_i$ is the population of $i$'s household.

## 3. Results

### 3.1. Description of the Statistical Population

The results showed that out of 371 respondents, the heads of 279 households were men, and 92 head households were women. Most heads of households (190) are illiterate. 6.5% of heads of households (24 people) were less than 30 years old, and 51.1% (93 people)

were over 60 years old. The highest frequency of respondents belonged to the age group 51–60 years. Meanwhile, 205 heads of households were currently employed, and 166 of them were unemployed (Figure 2).

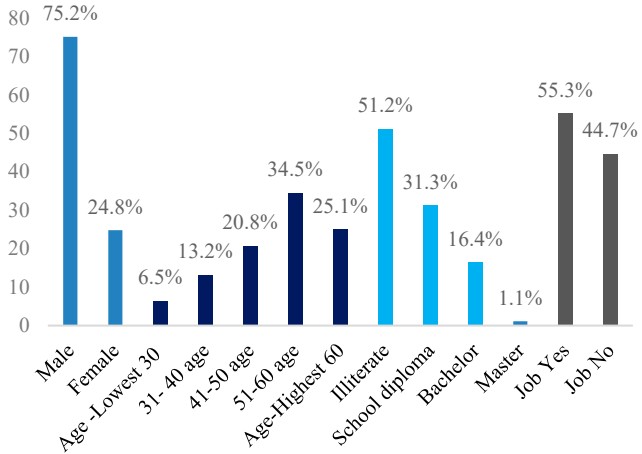

**Figure 2.** Characteristics of sex, age, education, and employment of all respondents (in percent).

Among the heads of rural households, 52.8% (Figure 3) of the respondents were illiterate, while this rate for the urban households was 44.9% (Figure 4). Like the illiteracy rate, the unemployment rate was higher among rural respondents (50.6%) compared to urban respondents (37.7%).

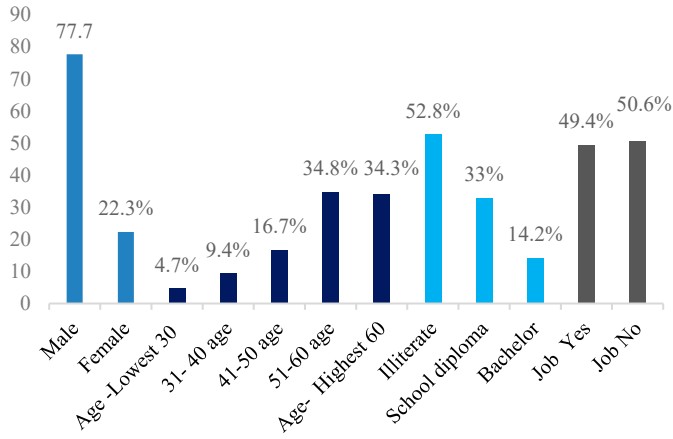

**Figure 3.** Characteristics of sex, age, education, and employment of rural households (in percent).

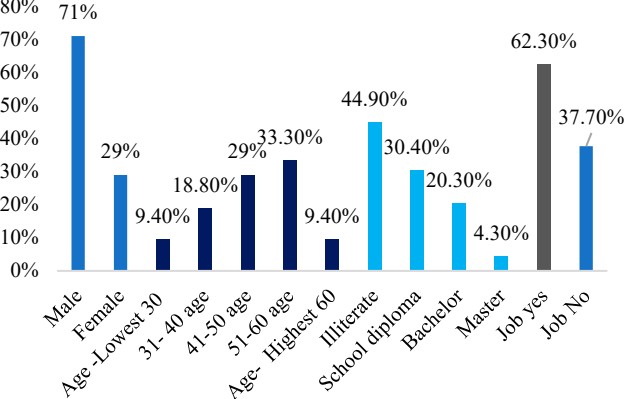

**Figure 4.** Characteristics of sex, age, education, and employment of urban households (in percent).

*3.2. The Intensity of Poverty (A)*

According to the study's findings, the province of Kunduz has an intensity of poverty index (A) value of 0.526. This indicates that the province's urban and rural households have lost access to 52.6% of educational, health, and welfare (living standard) facilities. However, the intensity of poverty in the rural and urban communities was 0.527 and 0.497, respectively. In other words, the value of the intensity of rural poverty is higher than in urban society, and the rural community is more deprived. In 56% of rural households, there were people aged 10 or older who were illiterate, and 75% of rural households reported having a child in their family between the ages of 6 and 10 who was not in school. 44% of the households had a woman who had died due to birth complications in the last 5 years, and 88% of rural households did not have access to the internet due to poverty. These data show the glaring intensity of poverty in rural areas (Figure 5).

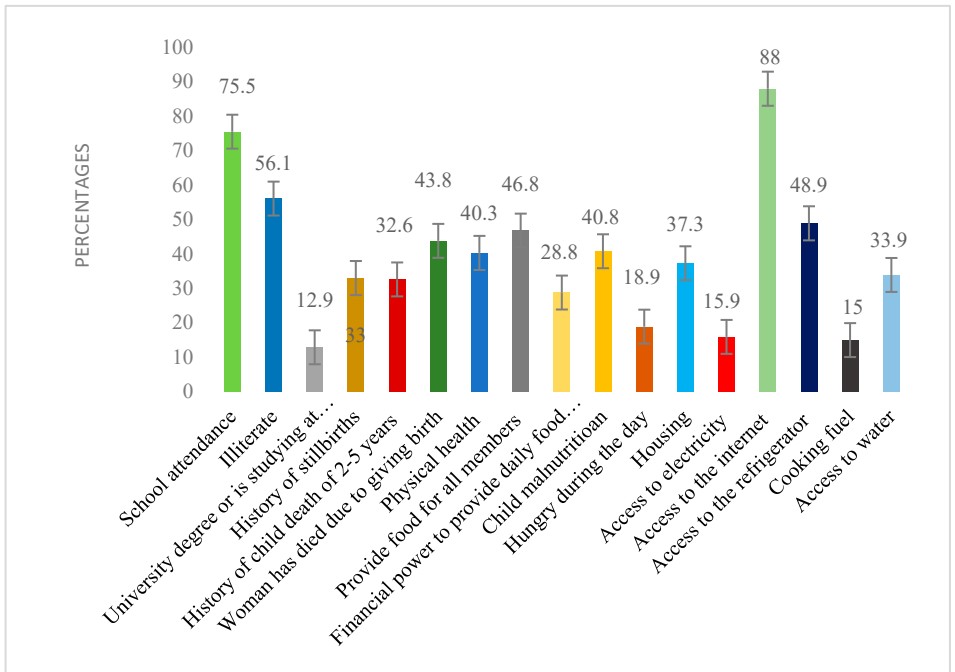

**Figure 5.** The intensity of poverty (A) in rural households (in percent).

Urban households, however, did not do much better in terms of the mentioned indicators (Figure 6). Only a few indicators showed that urban households were superior to rural ones and included access to electricity, heating of the house and cooking, and nutrition. Undoubtedly, it appears that urban households in this province now have better and more access to a variety of facilities and services due to reasons such as having a more suitable infrastructure and the government paying more attention to urban areas. Urban households, therefore, experience less poverty (Figure 6).

*3.3. The Headcount Ratio (H)*

The results showed that the headcount ratio of the poverty index (H) of the whole province was 0.849. However, comparing the H index between urban and rural areas shows that the urban households' value (0.895) was slightly lower than that of rural households (0.917). Although the value of H was very high in the province, it seems that rural households are in a worse situation than urban households in terms of the headcount ratio of the poverty index (Figure 7).

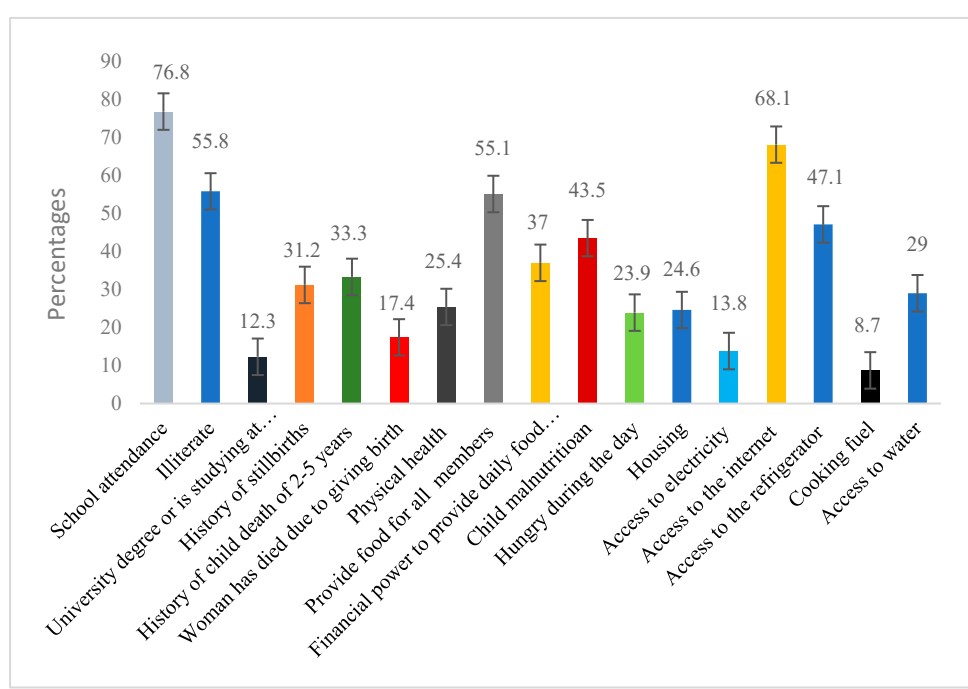

**Figure 6.** The intensity of poverty (A) in urban households (in percent).

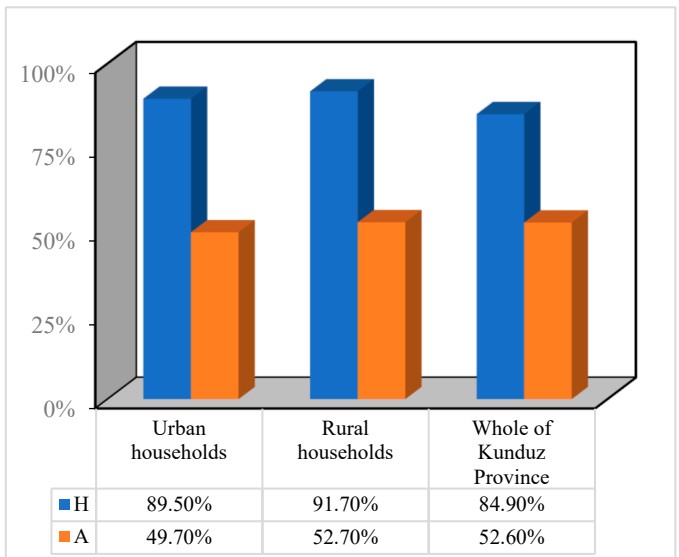

**Figure 7.** Headcount ratio index (H) in urban and rural households (in percent).

*3.4. Multidimensional Poverty and the Contribution of Its Dimensions*

The value of the MPI in the whole province was 0.446. This value for rural areas was 0.483, and for urban areas was 0.445. As previously mentioned, the MPI index represents a combination of the incident and intensity of poverty. It appears that rural households experienced greater concentrations of the MPI, such as poverty intensity, than urban households. Additionally, we found that the educational dimension, with 48.5%, had the highest contribution to the MPI. This contribution for urban and rural areas was 51.5% and 48.4%, respectively. Moreover, the share of the living standards dimension (27.2%) in the MPI for rural areas was more than the health dimension (24.4%). While for urban areas, the share of living standards dimension (23.4%) was less than the health dimension (25.1%). In other words, rural areas are faced with more deprivation in terms of living standards, but urban areas face more health and educational deprivation (Figure 8).

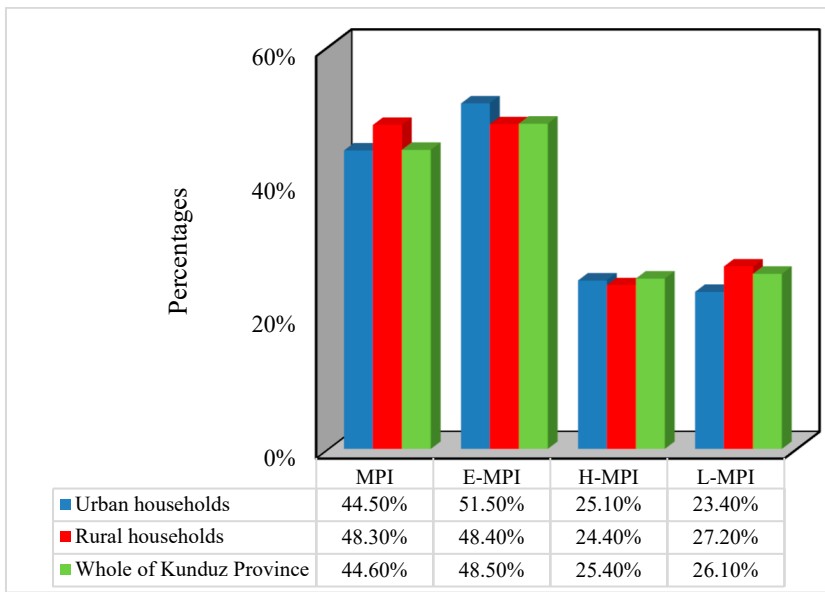

| | MPI | E-MPI | H-MPI | L-MPI |
|---|---|---|---|---|
| ■ Urban households | 44.50% | 51.50% | 25.10% | 23.40% |
| ■ Rural households | 48.30% | 48.40% | 24.40% | 27.20% |
| ■ Whole of Kunduz Province | 44.60% | 48.50% | 25.40% | 26.10% |

**Figure 8.** The MPI index and its dimensions' contribution.

## 4. Discussion

Poverty is neither an economic nor a one-dimensional phenomenon. In both rural and urban areas, a variety of dimensions and characteristics have been used to explain poverty, including education, household size, the age of the family's head of household, and availability of clean water [29–34].). In addition, poverty is a time- and place-oriented phenomenon, meaning that its character and severity vary in particular contexts. Thus, studying and analyzing poverty varies depending on the location and time, and it calls for specific indicators. As a result, we attempted to investigate poverty in the Kunduz province from various perspectives, including education, health, and welfare (living standards), as well as objective realities of poverty and indigenous indicators. Different dimensions of poverty were evaluated based on 17 indicators.

We found that the headcount ratio index of poverty (H) in the whole of the province was (84.9%), and H was higher than the intensity of poverty (A) in the province (both in rural and urban areas). Undoubtedly, the war and insecurity of the last two decades harmed the social and economic status of the Kunduz province. Despite the province's strong investment and private sector development prospects, it has not been suitable for private sector development due to recent instability. At the same time, the occupation of more than 80% of rural households in the Kunduz province is agriculture and animal husbandry. However, due to successive droughts, the agricultural sector has also suffered, leading to a lack of employment in rural areas. The population density in the center of the province and the cities appears to be at an all-time high. Therefore, the headcount ratio of poverty has increased not only in rural households and remote areas but also in the center of Kunduz and its cities. As the rural labor force migrates to the cities, unemployment increases due to a lack of employment opportunities in the cities. The lacking development of non-agricultural enterprises in rural regions, which is predominant in rural and urban areas, long-term drought, multiple wars, and other causes have all contributed to the expansion of poverty.

According to this study's findings, poverty in the Kunduz province appears to be more severe in rural and urban areas in terms of education rather than in terms of health and welfare. This demonstrates that poverty is not just an economic problem. There are various studies [21,23,35–37] that reported similar results around the world. Barati, Moradi, Zholideh, Sohrabi Mollayousef and Christine [24] found that poverty in Iranian villages emphasized education over health and living standards. Undoubtedly, an increased family size, along with factors such as government mismanagement, drought, and illiteracy, led to

rising multidimensional poverty in the Kunduz region. The family size in Kunduz was usually large; more than 61.7% of the population lived in households with 5–10 members, approximately 34% of them lived in households with 10 to 15 members, and about 2% of the population lived in families with more than 15 members. Concerning household structure, size, and economic condition, there is a high probability that large households had the largest contribution to MPI poverty. It seems that the MPI was also related to the education of head households. More than half of the province's population (51%) lived in households where the head of the household was illiterate. Hence, these populations have the highest occurrence of the MPI and the highest intensity of poverty.

The essential features of the MPI in rural regions included rural households' vulnerability to hazards and the resilience of the materials used in their homes, low-quality housing hygiene in terms of amenities, minimum basic infrastructure, and a lack of conventional fuel useable in homes. Investigating the MPI with the separation of urban and rural households showed that the value of the MPI was higher in rural households than in urban households. Hence, rural households faced more deprivation in terms of education than in the dimensions of health and living standards. Rural families appear to be more disadvantaged due to a lack of appropriate infrastructure in the field of education and living standards, as well as less government attention. Further, the education dimension of poverty was also more advanced at the level of urban households. Certainly, a lack of welfare and social services in rural regions has led to increased rural migration into cities, resulting in a rise in the MPI among urban households. However, urban households were in a better and more comfortable welfare situation than rural households in terms of living standards. In other words, the data indicate that the central government pays more attention to urban areas, and adequate welfare infrastructure is primarily located in cities.

Many studies have indicated that distance from the city center affected poverty and deprivation [38,39]. Aside from the geographical location, the climate is also one of the variables affecting rural poverty in the province of Kunduz. Various studies [40–42] have shown that being in unfavorable climatic regions had an undeniable role in the headcount ratio and intensity of poverty. So, natural disasters such as floods and droughts may wreak havoc on a country like Afghanistan, particularly in areas where the lack of financial resources to combat and recover from such occurrences is minimal. In the last few years, this phenomenon has often damaged the food resources of this province, and the residents of rural households have been faced with serious food shortages [43]. Therefore, in addition to the various socioeconomic challenges, urban households also feel food insecurity. Because agricultural activities are the primary income source for most Afghans, and these activities are vulnerable to climate change, the government should make adequate preparations, with the help of the international community and international organizations, to reduce the deadly effects of climate change.

The results of this study also showed that although Afghanistan has paid more attention to urban and rural development in recent years than ever before, the headcount ratio of poverty (H), the intensity of poverty (A), and the multidimensional poverty index (MPI) have not decreased significantly. As a result, regional orientation is linked to incorporating significant concerns into developing poverty-fighting programs, accurately recognizing the characteristics of disadvantaged groups, and implementing poverty-fighting methods (urban and rural). Because regions have differences in terms of the prevailing socioeconomic conditions, this necessitates a more detailed study among regions, especially in populous areas and areas where rural residents face various social, physical, and environmental deprivations. Most importantly, understanding the headcount ratio of poverty in society and distinguishing the impoverished from the prosperous at the community level is one of the most significant challenges in developing poverty reduction strategies. Given the susceptibility of villagers to income and price variations, the major measures to eliminate poverty should focus on empowering the rural poor through access to credit and banking facilities, participation in small rural credit funds, strengthening membership and participation in cooperative programs, access to job opportunities, implementation

and development of training, skills, technical and specialized considering of potential rural resources. The uncontrolled growth of urbanization in the last two decades, which has intensified in Kunduz province, has been partly due to rural poverty and the intensification of poverty. This is due to the lack of appropriate and inclusive employment and increased poverty in the rural community, which has harmed urban communities over time. If the poverty problem in this province, particularly in rural regions, is not addressed, a large-scale migration of villages to cities will occur, causing greater difficulties in cities and harming agricultural productivity.

The key motivation for designing the MPI for the Kunduz province was to guide evidence-based policies and programs that accelerate poverty reduction. For children, policies and interventions to improve their health, survival chances, education, and skills will affect their potential in the future and hence should be a priority. Improving food security, access to safe nutrition, maternal health, and school attendance will contribute the most to reducing the MPI. Because deprivations are interlinked, and children and families experience overlapping disadvantages, an emphasis on integrated multisectoral policies and programs is essential. As children are a particularly vulnerable group, with specific needs according to their age, there is a need to pay attention to children and their families in national budgets and invest equity in children. Accordingly, by taking into account the following advice, in particular, poverty can be decreased in this province:

- Review and apply the new strategies for reducing poverty according to the potential of the Kunduz province with the aims of structural change in the production system, elimination of the imbalance between urban and rural areas in the field of education, access to healthy water, establishment of rural markets, broadly shared economic growth in rural areas, and encouragement of public participation.
- While developing the activity of technical and vocational training centers to familiarize rural residents with technical and professional skills, develop small production workshops, and provide self-employment and self-sufficiency to rural households. According to the research findings, this is essential for mobility and acceleration in employment.
- Create a regional balance among public, social, and educational facilities in rural areas and pay more attention to the social, environmental, and human capabilities of the local areas and communities.
- Take into consideration offering mothers training sessions on a healthy and full diet with the aid of health facilities in order to decrease food insecurity and health poverty, especially among children and women who suffer from more deprivation.
- Diversify the sources of income and employment in rural regions by developing small, home-based businesses and extending the supply and value chains for main agricultural crops to include wheat, barley, rice, cotton, flax, mung beans, almonds, maize, and vegetables.

## 5. Conclusions

Afghanistan is one of the world's most vulnerable countries, experiencing many problems, including widespread poverty. Even though the issue of poverty has received much governmental and international attention and funding, it has worsened over time. Kunduz is one of the northeastern provinces of Afghanistan, with about 74% of the population living in rural areas and 26% living in urban areas. More people live in poverty in Kunduz than in many other places. Since poverty is not a one-dimensional phenomenon, this study evaluated it from different aspects based on specific criteria and indices. Additionally, it assessed the proportion of each dimension of poverty using regional indices in order to offer the relevant authorities specific recommendations for focused policies and the eradication of poverty.

The findings showed that (a) rural households in Kunduz province experienced greater poverty than urban households; the MPI for rural families was 0.483, with educational, living standards, and health having the most significant proportion of the MPI. As a result,

the Kunduz province is grappling with widespread rural and urban poverty, with rural regions experiencing greater severity than urban ones. (b) For urban and rural areas, the proportion of the three types of poverty varied. The MPI's educational component contributed the most to both regions, while in rural areas, the living standards component outweighed the health component. In contrast, the living standards in urban regions have lagged behind health. This indicates that whereas urban areas experience greater health and educational deprivations, rural areas experience greater living standard deprivation. (c) Finally, the intensity and incidence of poverty were not the same among rural and urban communities. Both indicators in rural communities are worse than in urban communities. This problem was influenced by a variety of circumstances, including unemployment, a lack of a coherent plan adapted to poverty reduction, recent droughts, illiteracy, and population increase in the province. In addition, our findings also showed that the household size was usually large in the Kunduz province (34% of respondents stated that they lived with 10–15 members of the family). Consequently, the rising household size was one of the contributors to the upsurge in multidimensional poverty. However, poverty was more concentrated in rural areas of the province. Therefore, poverty reduction policies should be based on empowering the rural poor with a focus on socioeconomic and climate change. Similarly, rural poverty and the intensity of poverty in these regions have contributed to the uncontrolled expansion of urbanization in the Kunduz province during the past two decades. While urban regions have a superior infrastructure to rural areas, a lack of job opportunities and a lack of focus on providing better services in rural areas is also contributing to the province's growing urban multidimensional poverty. Furthermore, if a specific plan is not adopted, particularly in the province's rural regions, we will confront mass migration to domestic and urban cities.

Although this study may have faults, it should be emphasized that there were more significant limitations than in previous studies. More than half of the data was gathered through surveys due to the prevalence of COVID-19 and a lack of internet connectivity. The government officials and municipal authorities in this province should undoubtedly be convinced and inspired to undertake creative poverty-reduction strategies as a result of this study.

**Author Contributions:** Conceptualization, M.A.S.; methodology, A.A.B.; software, M.A.S. and A.A.B.; validation, A.A.B. and K.K.; formal analysis, M.A.S. and A.A.B.; investigation, M.A.S. and A.A.B.; resources, M.A.S.; data curation, M.A.S.; writing—original draft preparation, M.A.S.; writing—review and editing, A.A.B.; visualization, A.A.B.; supervision, K.K. and A.A.; project administration, A.A. All authors have read and agreed to the published version of the manuscript.

**Funding:** This research received no external funding.

**Data Availability Statement:** Data will available if requested.

**Conflicts of Interest:** The authors declare no conflict of interest.

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
