# Peer review of "Dimensions of Poverty in Kunduz Province of Afghanistan"

_world, doi:10.3390/world3040055_

Round 1

Reviewer 1 Report

Good work, Congratulations The topic is very relevant and the results might be very useful for decision-makers. Minor editions and suggestions follow:

The following authors were not found in the reference list

Abdulai & Shamshiry, 2014 (row 50)
Central Statistics Organization, 2014 (rows 44 and 45)
Kambur, 1994 (row 50)
Poverty and Initiative, 2014 (rows 295-296)
Sen, 1999 (row 50)

In the reference list (pages 13 and 14) some are missing:

#13. (row 474)
#33. (row 509)
#37. (row 516)

Two small typos to correct:

Row 15 (Abstract): it reads "(...) but the rural poverty is severe than urban poverty (...)". I guess it should read: "but rural poverty is more severe than urban poverty" Shouldn´t be?

Row 112. There is an extra dot "." at the end of the sentence: Northeastern Afghanistan was the location of the study.."

Row 125. Remove the dot where it reads: "(...) grape. (MEA, 2019)" and add it to the end: "(...) grape (MEA, 2019)."

Table 1. The first indicator under the "Education" dimension: "No household member with the age of 6-10, doesn´t go to school (1/3)". There is a double negative. I think it should read: "No household member with the age of 6-10, does go to school (1/3)" or "No household member with the age of 6-10, goes to school (1/3)"

Rows 352-354, where it reads: "Given the susceptibility of villages to income and price variations, the major measures to eliminate poverty should focus on empowering the rural poor" What does it mean, exactly? What is the precise meaning of "empowering the rural poor"? Please explain it in a more clear and simple way. It sounds too vague.

Finally, as a more conceptual issue, You have been talking about poverty in Kunduz, and you mentioned some of the agricultural products that can be grown there. But haven´t said a word about what are the real sustainable possibilities for jobs and trade, for instance. It seems that besides education and health, everything has to be brought by the government. Are you implying that they cannot survive on their own and that Gov will have to maintain intervention forever for these communities to reach some level of development? I would like to see at least some brief reflection about that in the work.

Author Response

No

Comments

Response

Page(s)

1

Good work, Congratulations The topic is very relevant and the results might be very useful for decision-makers. Minor editions and suggestions follow:

The following authors were not found in the reference list

Abdulai & Shamshiry, 2014 (row 50)

Central Statistics Organization, 2014 (rows 44 and 45)

Kambur, 1994 (row 50)

Poverty and Initiative, 2014 (rows 295-296)

Sen, 1999 (row 50)

Modified

Thank you for your friendly comment. All references were checked and added to the list of references. Please see the references list.

Pages 15-16.

2

In the reference list (pages 13 and 14) some are missing:

#13. (row 474)

#33. (row 509)

#37. (row 516)

Modified

 Thank you for your careful comment. All errors in the reference list have been corrected. Please see the references list.

Pages 15-16

3

Two small typos to correct:

Row 15 (Abstract): it reads "(...) but the rural poverty is severe than urban poverty (...)". I guess it should read: "but rural poverty is more severe than urban poverty" Shouldn´t be?

Modified

Thank you for your suggestion. The sentences corrected as your good comment. Please see the references list.

Page 1

4

Row 112. There is an extra dot "." at the end of the sentence: Northeastern Afghanistan was the location of the study.."

Modified

Thank you for your attention. The extra dot was removed.

Page 3

5

Row 125. Remove the dot where it reads: "(...) grape. (MEA, 2019)" and add it to the end: "(...) grape (MEA, 2019)."

Modified

Thank you for your attention. The extra dot was removed.

Page 4

6

Table 1. The first indicator under the "Education" dimension: "No household member with the age of 6-10, doesn´t go to school (1/3)". There is a double negative. I think it should read: "No household member with the age of 6-10, does go to school (1/3)" or "No household member with the age of 6-10, goes to school (1/3)"

Modified

Thank you for your attention. You are right these indicators were modified as follow:

"No household member with the age of 6-10, goes to school (1/3)"

Page 5

Rows 352-354, where it reads: "Given the susceptibility of villages to income and price variations, the major measures to eliminate poverty should focus on empowering the rural poor" What does it mean, exactly? What is the precise meaning of "empowering the rural poor"? Please explain it in a more clear and simple way. It sounds too vague.

Explained

Thank you for your comment. We modified and explained these sentences as follow:

" Given the susceptibility of villagers to income and price variations, the major measures to eliminate poverty should focus on empowering the rural poor through the access of the rural poor to credit and banking facilities, their participation in small rural credit funds, strengthening the membership and participation of the rural poor in cooperative programs, the access of the rural poor to job opportunities, implementation and development of training, skills, technical and specialized considering of potential rural resources. "

Page 13

Finally, as a more conceptual issue, You have been talking about poverty in Kunduz, and you mentioned some of the agricultural products that can be grown there. But haven´t said a word about what are the real sustainable possibilities for jobs and trade, for instance. It seems that besides education and health, everything has to be brought by the government. Are you implying that they cannot survive on their own and that Gov will have to maintain intervention forever for these communities to reach some level of development? I would like to see at least some brief reflection about that in the work.

Explained

Thank you for your valuable comment. Although the authors completely agree with your point, as we explained on page 11, " Undoubtedly, the war and insecurity of the last two decades have had a negative impact on the social- economic status of Kunduz province. Despite the province's strong investment prospects and private sector development, it has not been a suitable stand for the private sector due to recent instability."

As a result, first of all, peace and tranquility and appropriate infrastructures should be provided in the study area, so that people and the private sector will be more willing to participate in the development process. In other words, people must first make sure that the government has enough determination to support the participation and activity of the private sector. It is in this direction that the main emphasis of this article was on the need for more government activity. We have added explanations in this regard to the introduction, discussion and conclusion sections to clarify this issue.

Pages 2, 11-14

Please see the attached file for a better view.

Reviewer 2 Report

I think my comments come down to a much clearer line of argument (no repetitions, what is the question and what is the answer, repeated reference to MPI to different articles, no new material in the conclusions, no "it has been said" in the conclusions). The content of the article deserves a thorough re-reading and subsequent re-writing.   

Could you please look at the references? No 'italics'? Repetitions? I don't understand reference 13. 33. 37.

Author Response

No

Comments

Response

Page(s)

1

I think my comments come down to a much clearer line of argument (no repetitions, what is the question and what is the answer, repeated reference to MPI to different articles, no new material in the conclusions, no "it has been said" in the conclusions). The content of the article deserves a thorough re-reading and subsequent re-writing.  

Explained

 We mentioned the main research questions at the end of the introduction section, and the article actually sought answers to these questions. We tried to provide clear answers to these questions in the summary section.

Also, the reason for using repeated indicators was that the results can be compared with other results in the world. Apart from this, unfortunately, due to the social, cultural and political conditions affecting the country of Afghanistan, it was very difficult and almost impossible to collect more detailed data, and for this reason, the present study was limited to examining these indicators.

We tried to solve the English problems of the text once again carefully read the text from the beginning to the end and fix the problems as much as possible.

We hope that the corrections made could increase the quality of the article.

-

2

Could you please look at the references? No 'italics'? Repetitions? I don't understand reference 13. 33. 37.

Modified

 Thank you for your careful comment. Unfortunately, in the process of transferring the contents to the Journal template, some references had errors. All errors in the reference list have been corrected. Please see the references list.

Pages 15-16.

Please see the attached file for a better view.

Reviewer 3 Report

It's an interesting research topic that can attend global readership. The research gap is clear and the data set is good and the analysis and its interpretations are acceptable to reach a conclusion. However, authors should insert separate sections to present the implication for policymaking.  The conclusion should be more accurate and precise.  

Author Response

No

Comments

Response

Page(s)

1

It's an interesting research topic that can attend global readership. The research gap is clear and the data set is good and the analysis and its interpretations are acceptable to reach a conclusion. However, authors should insert separate sections to present the implication for policymaking.  The conclusion should be more accurate and precise. 

Modified

Thank you for your friendly comment. The entire manuscript and especially the conclusion section was revised and rewritten to provide more assistance to policy makers and planners for poverty management and reduction.

Pages 15-16.

Please see the attached file for a better view.

Round 2

Reviewer 2 Report

minor corrections: line 87 'deprived of ... of poverty' should be 'suffered from ... of poverty'

line 113: reformulate first sentence: 'The study area was Kunduz, a province located ...'

line 392: 'moms' should be 'mothers'

Author Response

Comment: line 87 'deprived of ... of poverty' should be 'suffered from ... of poverty'

Author's Response: Thank you for your comment. This sentence was modified as follows:

"Afghanistan were suffered from at least one dimension of poverty"

Comment: line 113: reformulate first sentence: 'The study area was Kunduz, a province located ...'

Author's Response: Thank you for your comment. This sentence was modified as follows:

"The study area was Kunduz, a province located in northeastern Afghanistan."

Comment: line 392: 'moms' should be 'mothers'

Author's Response: Thank you for your comment. The word "moms" was replaced by "mothers".
